# LeanComb: A Combinatorial Identities Benchmark for Theorem Proving via Automated Theorem Generation

## Abstract

Automated theorem proving (ATP) in complex mathematical domains remains a fundamental challenge for large language models (LLMs), due to the scarcity and imbalance of formalized training data. Combinatorics, with its discrete structures and symbolic reasoning, provides a demanding testbed for evaluating ATP capabilities. Addressing this data scarcity gap, we propose a comprehensive data-centric framework built upon two essential components: LeanComb, a high-quality human-curated dataset, and ATG4CI, a novel method for automated theorem generation. LeanComb is a manually curated dataset of formalized combinatorial identities in Lean 4. It encompasses eight fundamental areas of combinatorics, with training and test sets derived from the classical literature, enabling robust evaluation of cross-domain generalization. To overcome the data sparsity, we develop a data augmentation framework, the **A**utomated **T**heorem **G**enerator for **C**ombinatorial **I**dentities (ATG4CI). It introduces a novel "Learn-from-Failure" paradigm, combining LLM-guided exploration with reinforcement learning-driven search to systematically discover new theorems from the boundaries of models' reasoning capabilities. Applied to LeanComb, ATG4CI generates over 260K Lean-verifiable theorems, each with a complete proof. Fine-tuning models on the human-curated training set and the augmented dataset results in average improvements of $4.0\%$ and $7.2\%$, respectively, on LeanComb-Test set. The fine-tuned models also achieve promising performance on challenging ATP benchmarks, PutnamBench and CombiBench, demonstrating the effectiveness of our approach.

## 1 Introduction

Large language models (LLMs) have demonstrated impressive performance on well-structured automated theorem proving (ATP) tasks (Wei et al., 2024; Xin et al., 2025; Lin et al., 2025; Guo et al., 2025; Wang et al., 2025). However, their capabilities remain limited in more complex mathematical domains, particularly due to the scarcity and imbalance of formalized training data (Wei et al., 2024; Yu et al., 2025). These limitations are especially pronounced in fields that demand intricate symbolic reasoning. Combinatorial identities are central to the study of combinatorics (Britz, 2010), and play an important role in a wide range of mathematical and computational disciplines, including algebra, probability, and algorithm design (Chen & Guo, 2024; Konvalinka, 2008). They serve as essential tools for counting, enumeration, and establishing relationships between discrete structures. Despite their fundamental importance, automated theorem proving for combinatorial identities remains highly challenging due to the inherently discrete and structural nature of combinatorics, as well as the complexity and length of intricate proofs (Trinh et al., 2024; LessWrong, 2024).

These challenges necessitate large-scale training data, which is severely lacking in current formal theorem libraries. For instance, although the Lean-based standard library Mathlib4 (The mathlib Community, 2020) contains over 213K theorems and proofs, it includes fewer than 2K results related to combinatorics, with only around 100 specifically involving combinatorial identities. This stark imbalance highlights the urgent need for domain-specific formal data. While various automated theorem generation (ATG) approaches have been proposed to alleviate data scarcity, these methods generally focus on broad or well-studied domains. INT (Wu et al., 2021) and STP (Dong & Ma, 2025)

synthesize theorems from axioms or conjectures, while LeanDojo (Yang et al., 2024), Alchemy (Wu et al., 2024a), and MUSTARD (Huang et al., 2024) extract or generate formal statements by applying known tactics within existing formal mathematics libraries (Wu et al., 2024a; Huang et al., 2024). However, automated theorem generation for combinatorial identities remains largely unexplored by existing techniques, creating a critical shortfall in this specialized area.

To mitigate this shortfall, we propose a data-centric solution by constructing a specialized formalized dataset and developing an automated theorem generation framework. Specifically, we introduce LEANCOMB, a manually curated dataset of formalized combinatorial identities in Lean 4 (Ying et al., 2024). Covering eight core topics, LEANCOMB comprises a training set of $418$ combinatorial identities ( along with $209$ supporting lemmas), each accompanied by formal statements and proofs, as well as a test set of $100$ combinatorial identities containing only formal statements. To further augment the training data, we propose ATG4CI, a self-improving framework of automated theorem generation for combinatorial identities. This framework integrates LLM-guided tactic exploration with reinforcement learning-driven Monte Carlo Tree Search (MCTS) to generate high-quality theorems. Using the training set of LEANCOMB, this framework builds a large-scale dataset, LEANCOMB++, containing $260,466$ Lean-verifiable theorems with full proofs. We evaluate models fine-tuned on both LEANCOMB and LEANCOMB++. Our models achieve up to 25% pass@8 accuracy on the test set, with average gains of 4.0% and 7.2% from the human-curated and generated data, respectively. Moreover, our fine-tuned models exhibit generalization on challenging formal benchmarks, including PutnamBench (Tsoukalas et al., 2024) and CombiBench (Liu et al., 2025), demonstrating the utility of our datasets and approach for advancing automated theorem proving in combinatorics. Our contributions are summarized as follows:

- We present LEANCOMB, a human-annotated dataset of formalized combinatorial identities in Lean 4, covering 8 core areas. The training and test sets are drawn from different classical references, enabling robust evaluation of generalization.

- We propose ATG4CI, a novel framework that transforms failed proof attempts into valuable training signals. It systematically generates new, verifiable theorems from the boundaries of a model's current reasoning capabilities, establishing a self-improving paradigm for data creation in specialized mathematical domains. It generates LEANCOMB++, a large dataset of 260 K+ formally verified combinatorial theorems.

- We show that models fine-tuned on LEANCOMB and LEANCOMB++ achieve significant improvements on our test set and promising performance on two external ATP benchmarks, highlighting the domain-specific value and general applicability of our framework.

## 2 RELATED WORK

**Automated Theorem Proving and Generation.** Mathematical reasoning has gained increasing attention in artificial intelligence (Saxton et al., 2019; Wang et al., 2023). ATP systems typically follow two main paradigms: tree search and whole-proof generation. Tree search methods, like GPT-f (Polu & Sutskever, 2020), explore proof steps incrementally, while BFS-Prover (Xin et al., 2025) scales this approach with breadth-first search. MPS-Prover (Liang et al., 2025) creates structured plans to guide subsequent formalization. Whole-proof generation methods synthesize complete proofs in a single step. Leading representatives of this approach, such as Goedel-Prover-V2 (Lin et al., 2025), DeepSeek-Prover-v2 (Guo et al., 2025), Kimina-Prover (Wang et al., 2025), and Seed-Prover (Chen et al., 2025) incorporate expert iteration, massive synthetic data, reinforcement learning, and self-exploration, achieving state-of-the-art results on general benchmark. A key challenge fueling ATP research is the acquisition of training data, which has led to the growing subfield of automated theorem generation (ATG). Recent work increasingly employs LLMs for this task. MUSTARD (Huang et al., 2024) uses a concept-driven pipeline to produce high-quality theorem–proof pairs, while AIPS (Wei et al., 2024) enables proof generation for algebraic inequalities without human demonstrations. Extending beyond static datasets, STP (Dong & Ma, 2025) proposes a self-play framework where conjecturing and proving co-evolve. Despite these advances, both general ATP and specialized ATG methods remain limited in domains like combinatorics, motivating our work on a scalable generation framework for this area.

**Datasets and Benchmarks for Theorem Proving.** Lean's mathematical library, Mathlib (The mathlib Community, 2020), serves as the primary source of training data for many neural theorem

provers, containing a large collection of formalized theorems. To further enhance this, datasets like HERALD have been generated via data augmentation. Another large-scale resource is Lean WorkBook (Ying et al., 2024), which provides formal proofs sourced from mathematics competitions. For evaluation, several benchmarks have been proposed. MiniF2F (Zheng et al., 2022) and PuttAna target problems from high-level competitions, while FormalMATH (Yu et al., 2025) offers a broad, domain-diverse testbed. More specialized benchmarks include CombiBench (Liu et al., 2025) for combinatorics, FIMO (Liu et al., 2023) for IMO problems, and ProofNet (Geuvers, 2009) for undergraduate-level mathematics. However, a common limitation across these resources is the scarcity of combinatorial data.

## 3 LEANCOMB: A COMBINATORIAL IDENTITIES DATASET

We introduce LEANCOMB, a manually curated dataset of formalized combinatorial identities in Lean 4, designed to support ATP research in symbolic mathematics. Unlike prior datasets, LEANCOMB focuses on identities that require precise algebraic manipulation and multi-step reasoning. The dataset is split into 627 training theorems, each equipped with full Lean 4 formal proofs, and 100 test theorems, which are proof-free and drawn from broader combinatorics literature (Gould, 1972), enabling evaluation under full synthesis settings. The training identities are sourced from classical references (Spivey, 2019; Shi, 2001). To ensure both novelty and coverage, any theorems overlapping with Mathlib4, PutnamBench, CombiBench, FIMO, and ProofNet were rigorously removed. The construction process involved more than 1,800 hours of expert effort from a team of 15+ trained formalizers, each following a detailed annotation protocol. Further dataset statistics and curation methodology are available in Appendix B.

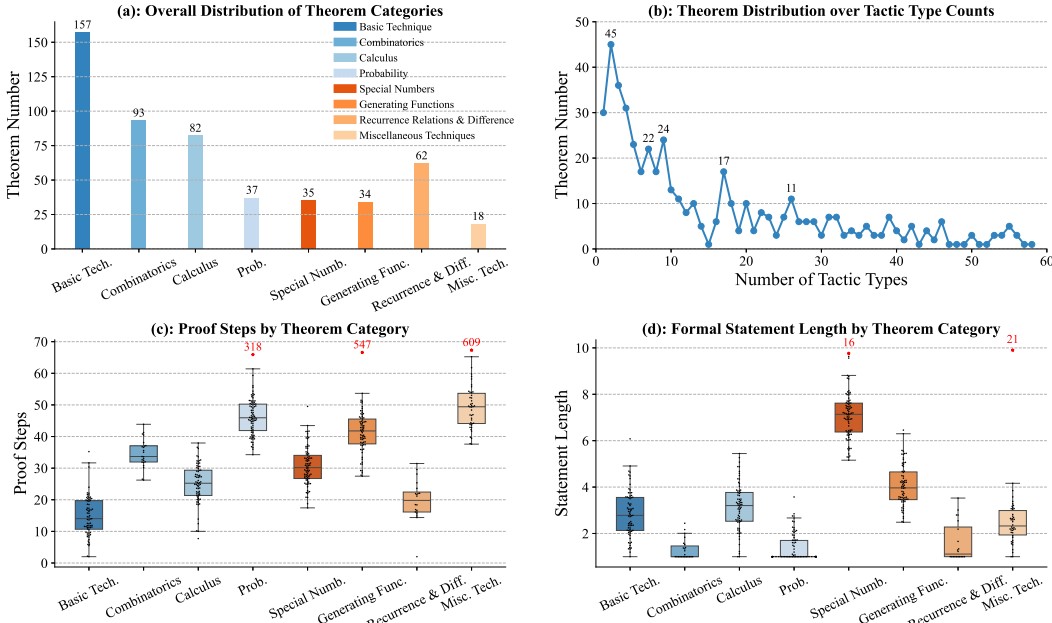

Figure 1: **Statistics of the LEANCOMB dataset.** (a) *Theorem categories*: the dataset spans 8 mathematical areas, with the majority coming from Basic Techniques (157), Combinatorics (93), and Calculus (82). (b) *Tactic diversity*: most proofs require 2–6 unique tactic types, with some using over 60; the maximum reaches 282, indicating high procedural complexity. (c) *Proof length*: average proof lengths range from 33–38 steps, depending on category, with some extreme cases exceeding 400 steps. (d) *Statement complexity*: formal statements vary in syntactic length across topics, with Special Numbers and Generating Functions often exhibiting the highest structural depth.

Figure 1 presents key dataset characteristics: (a) Theorem Categories. Most entries fall under Basic Techniques (157), followed by Combinatorics (93). Categories such as Generating Functions (34) are relatively underrepresented, reflecting both their mathematical niche and the scarcity of formal resources in these areas. (b) Tactic Diversity. While the majority of theorems use between 2 and 6

different tactic types, 21.7% involve more than 60 distinct tactics, with the most complex proof using 282 unique tactics. This highlights the need for ATP systems to master not just individual tactics but also strategy composition across reasoning chains. (c) Proof Length. Proof depths vary significantly by topic. Average lengths are 38 steps for Basic Techniques, 35 for Combinatorics, and 33 for Calculus. Several outliers, especially in Generating Functions and Probability, require over 400 steps (e.g., 509 steps in one instance), posing a substantial challenge for both annotation and automated reasoning. (d) Statement Complexity. We measure the syntactic complexity of theorem statements by code length. Categories with sparse mathematical definitions—such as Special Numbers and Generating Functions—tend to produce longer, more nested statements. In contrast, Combinatorics typically benefits from richer existing definitions, resulting in relatively concise formal encodings.

## 4 THE FRAMEWORK OF AUTOMATED THEOREM GENERATION

This section introduces ATG4CI, a general iterative framework for automated theorem generation designed to enhance the discovery of new theorems through policy-guided prediction. In this work, ATG4CI is built upon LEANCOMB training set, combining self-improving LLMs with a reinforcement learning-based search algorithm. Candidate theorems are generated via policy prediction and filtered through a theorem verification process, with only non-redundant and correct theorems retained to construct the extended LEANCOMB++ dataset.

Shown in Fig. 2, ATG4CI comprises three stages: *Partial Proof Paths (P3s) Construction*, *Candidate Theorem Generation*, and *Theorem Validation*. The procedure begins with the LEANCOMB training set $L_t^*$, consisting of formalized identities, serving as the foundational data for data augmentation and model fine-tuning. Using this, the pipeline enters the *P3s Construction* phase, where Lean4Kit is employed to transform each theorem into a proof tree. Their corresponding state-policy pairs are extracted from these trees. The states are then corrected to ensure their quality and correctness.

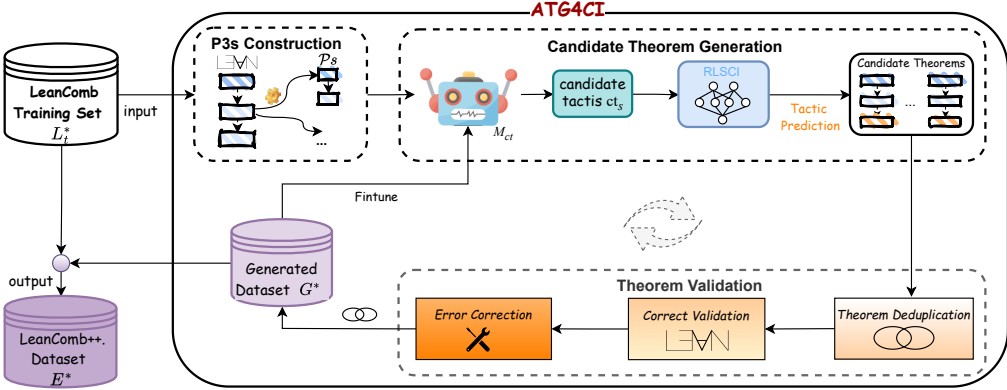

Figure 2: The Framework of ATG4CI.

Following this, in the *Candidate Theorem Generation* phase (top right of Fig. 2), the procedure begins with generating candidate tactics using a fine-tuned model $M_{ct}$. The candidate tactics are then refined by selecting the most appropriate ones for each partial proof path (*P3*) through a Reinforcement Learning-based search tailored to address the specific requirements of the combinatorial identities domain. This tactic prediction process is repeated to ultimately derive candidate theorems, followed by the *Theorem Validation* stage (bottom of Fig. 2), where redundant theorems are eliminated, and the correctness of the theorems is verified. Specifically, this stage consists of two key steps: *Theorem Deduplication* and *Error Correction*, both of which ensure the uniqueness and correctness of the dataset. Ultimately, the validated theorems are compiled into a new dataset $G^*$, fed to train the model $M_{ct}$, enhancing its ability to generate more effective candidate tactics. Finally, the generated dataset is combined with the LEANCOMB training set $L_t^*$ to form the LEANCOMB++ dataset $E^*$, which subsequently improves the performance of automated theorem proving. We demonstrate the procedure of ATG4CI using a typical example is provided in Appendix A.

### 4.1 PARTIAL PROOF PATHS (*P3s*) CONSTRUCTION

We focus on the *P3s* construction process, explaining how to use our Lean4Kit tool to extract partial proof paths $\mathcal{P}_s$ from formalized theorems. Built on Lean 4, Lean4Kit supports robust data extraction and interaction with both Lean and LLMs, with further details available in Appendix D. To facilitate the exploration of new proof paths for *P3s*, the tool visualizes all tactics within the proof environment, capturing the state transitions that occur before and after the application of each tactic. A fully formalized proven theorem, consisting of $n$ tactics, can be represented as a proof tree:

the root theorem forms the root node, tactics are the edges, intermediate states are the child nodes, and the "*no goals*" state is the leaf node. From this tree, $\mathcal{P}_s = \{p_i\}_{i=1}^{n-1}$ is extracted by tracing the paths from the root to the intermediate states. For example, the root theorem shown in Fig. 3, represented by the root node in $\sum_{k=1}^{n} nC_{n-1}^{k-1} = n \sum_{l=0}^{n-1} C_{n-1}^{l}$, transits through three sequential tactics and eventually reaches the "no goals" state at the leaf node, resulting in a four-layer proof tree. From this tree, two *P3s* are extracted: one from the root to the state after applying "$rw[mul\_sum]$", and another from the root to the state after applying "$rw[sum\_Ico\_eq\_sum\_range]$".

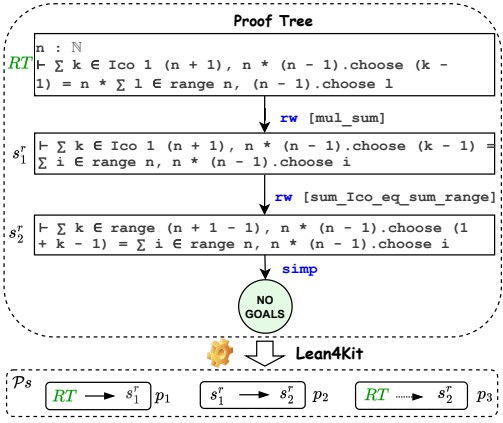

Figure 3: *P3s* Constructed by Lean4Kit.

### 4.2 CANDIDATE THEOREM GENERATION

The section explains how to generate candidate theorems from a given partial proof path, which consists of two key steps: candidate tactic generation and tactic prediction, based on a reinforcement learning search for combinatorial identities (RLSCI). We start with employing a fine-tuned model $M_{ct}$, trained by the training set of LEANCOMB, to generate candidate tactics $ct_s$ for *P3s*.

**Candidate Tactic Generation.** To enhance the quality and diversity of candidate tactics, we adopt an iterative refinement tactic inspired by self-improving techniques for fine-tuning models. Initially, the model is fine-tuned on the training set of the LEANCOMB dataset $L_t^*$ and Lean's foundational library, Mathlib4. Given a partial proof path, the improved model can generate $n$ candidate tactics, where $n$ is a prior positive integer. The model will be continuously refined in subsequent iterations through fine-tuning with the augmented theorems generated from the previous iteration. This iterative framework not only enhances the model's capacity to propose effective tactics but also broadens its exploration of diverse tactic spaces.

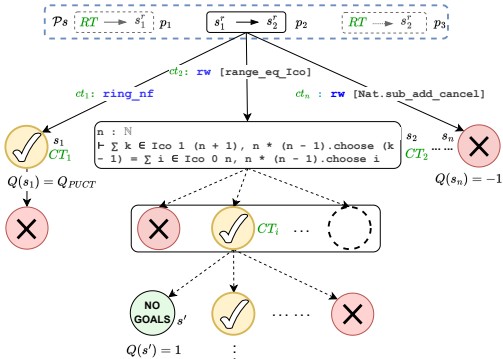

Figure 4: Candidate Theorems with RLSCI.

**Tactic Prediction.** The tactic prediction, based on RLSCI, consists of three primary steps: selecting candidate tactics, expanding the *P3s*, and back-propagating values from the leaf nodes to the root. These steps are iteratively performed until the proof is completed or no viable tactic is identified. If successful, the process discovers a complete proof path for the root theorem $RT$; otherwise, it generates a candidate proof path $cp_k$ for $k = 0, 1, \ldots, s$. Our RL framework comprises a critic model $C_\theta$ and a policy model $P_\theta$. Completed proof nodes are assigned a value of 1, while failed nodes are assigned a value of $-1$. Unresolved nodes are evaluated using the Polynomial Upper Confidence Trees (PUCT) method (Silver et al., 2017):

$$Q_{PUCT}(s) = Q(s, t) + c_{puct} \cdot P(s, t) \cdot \frac{\sqrt{\sum_b N(s, b)}}{N(s, t) + 1},$$

where $Q(s, t)$ is the estimated value of the state-tactic pair, obtained from the value network or learned from past simulations, and $P(s, t)$ is the probability of selecting a tactic in state $s$ based

on the policy network, and $c_{puct}$ is the exploration coefficient. In contrast, $N(s,t)$ is the count of executions of tactic $t$ in state $s$ during the prediction process. All successful tactics are stored as training data for LLMs. During backpropagation, values from the leaf nodes are propagated to the root, updating the visit counts $N(s,t)$ and cumulative action values.

The process aims to propagate proof paths as far as possible, terminating when the proof is complete or when no viable tactic can be found. In the former case, the leaf node is marked as "no goals", indicating the discovery of a new proof for the root theorem $RT$. In the latter case, the goals corresponding to the leaf nodes of the candidate proof paths are treated as candidate theorems, $CT_m = \{CT_i\}_{i=0}^m$. To generate new proofs, we incorporate the original root theorem $RT$ into the hypotheses and then apply tactics from the candidate path until the goal aligns with the target. Once aligned, the "assumption" tactic resolves the goal, completing the proof.

### 4.3 THEOREM VALIDATION

Note that not all candidate theorems are correct or unique, so a validation process, including theorem deduplication and correction, is necessary to retain the valid theorems. A detailed description of theorem validation is given in Appendix C.

**Theorem Deduplication**: Candidate theorems are deduplicated from two perspectives. First, textual duplication is identified by comparing the goals, premises, and proof steps. Identical theorems are merged, retaining only one. Second, mathematical equivalence is checked by simplifying redundant terms (e.g., $+0$, $-0$, $*1$, and $/1$), ensuring that only one mathematically equivalent version is kept.

**Theorem Correction**: After deduplication, as some candidate theorems may still contain expression errors or fail to pass the proof process, correctness refinement is performed. After candidate theorems are verified successfully by interacting with Lean, they are directly added to the generated dataset $G^*$. Those that fail verification are categorized by error type and corrected using the corresponding correction methods. The corrected theorems are then added to $G^*$.

We introduce the main steps of ATG4CI implemented in Algorithm 1. The procedure takes as inputs the training set of LEANCOMB dataset $L_t^*$, the model $M_{ct}$ that provides candidate tactics $ct_s$, search method RLSCI, the maximum round of iterations $n$, and returns the LEANCOMB++ dataset $E^*$. We construct partial proof paths $\mathcal{P}_s$ and initialize the generated dataset as $G_0$ based on the LEANCOMB training set $L_t^*$. Subsequently, the following steps are iteratively performed within a predefined maximum number of iterations. First, the model $M_{ct}$ is fine-tuned using the current generated dataset (as described in Line 4). Next, based on *P3s*, $M_{ct}$ is employed to generate candidate tactics (line 5). Subsequently, the search algorithm RLSCI is applied to generate candidate theorems (Line 6). The generated candidate theorems must undergo rigorous validation before being incorporated into the generated dataset $G_i^*$ and further integrated into the enhanced dataset $E^*$ (Line 7-8).

---

**Algorithm 1** The Framework of ATG4CI

---

**Input:** the training set of LEANCOMB Dataset $L_t^*$, the model $M_{ct}$ that provides candidate tactics
    $ct_s$, search method RLSCI, maximum round of iterations $n$
**Output:** LEANCOMB++ dataset $\mathbf{E}^*$
  1: $\mathcal{P}_s \leftarrow Construct\_P3s(L_t^*)$
  2: $G_0 \leftarrow L_t^*$
  3: $E^* \leftarrow L_t^*$
  4: **for** $i = 0 \rightarrow n$ **do**
  5:    $M_{ct}^i \leftarrow Fintune(G_i)$
  6:    $ct_s \leftarrow Produce\_ct_s(\mathcal{P}_s, M_{ct}^i)$
  7:    $G_i \leftarrow Generate\_CT_s(ct_s, RLSCI)$
  8:    $G_i^* \leftarrow Validation(G_i)$
  9:    $E^* \leftarrow E^* + G_i^*$
10: **end for**
11: **return** $\mathbf{E}^*$

---

## 5 EXPERIMENTS

This section evaluates the effectiveness and generalization of our data-centric framework. LLMs are fine-tuned on both the handcrafted dataset and the generated one. To evaluate the effectiveness of our datasets, we report pass@8 on the LEANCOMB-Test set, and two increasingly challenging external benchmarks: the mathematics competition dataset PutnamBench and the recently proposed combinatorics-focused benchmark CombiBench. Experimental results across these benchmarks demonstrate the effectiveness and generalization of the approach.

### 5.1 MAIN RESULTS

**Analysis of LEANCOMB++ Dataset.** Using the training set of LEANCOMB as a seed dataset, our generator, ATG4CI, generate 260,466 novel combinatorics theorems in two iterations, forming the enhanced dataset, LEANCOMB++. To provide candidate tactics for RLSCI, we iteratively fine-tune a general-purpose LLM, LLaMA3.1-8B (Biderman et al., 2023) with generated data in each round.

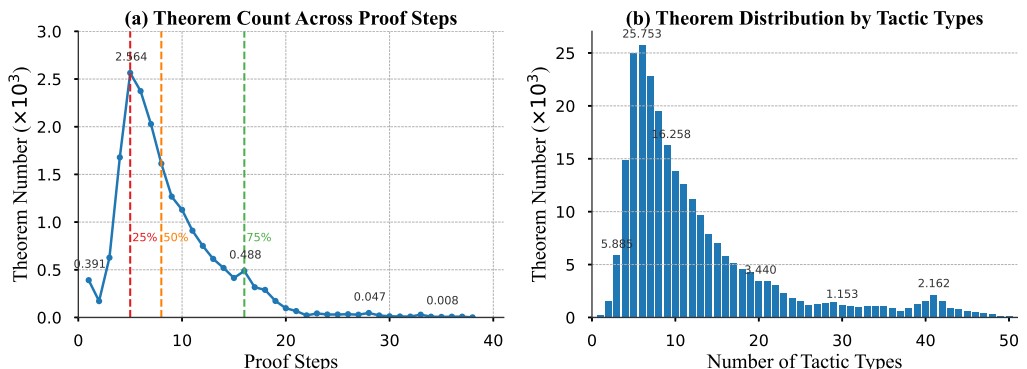

Figure 5: Theorem Distribution by Proof Steps and Tactic Types in LEANCOMB++.

To assess the difficulty and quality of LEANCOMB++, we examine the distribution of proof steps and tactic types. Figure 5(a) shows that while a significant portion of proofs are short (50% are within 8 steps), a substantial long tail of 65,117 theorems requires more than 16 steps, with the longest proof reaching 192 steps. For tactic diversity (Figure 5(b)), while most proofs use a small set of common tactics, 5,857 theorems require more than 20 distinct tactics, indicating high procedural complexity. This long-tail distribution indicates that ATG4CI successfully discovers a significant number of high-complexity theorems beyond the scope of trivial proof search, which is crucial for training models to move beyond simple pattern matching and develop long-horizon reasoning capabilities.

**Model Performance on LEANCOMB-Test.** We evaluate the impact of fine-tuning on our datasets with respect to in-domain performance. As shown in Table 1, training on the hand-curated LEAN-COMB dataset results in performance gains of up to 6 percentage points, with models like DeepSeek-Prover-v2 (Guo et al., 2025) reaching 25% pass@8. Training on LEANCOMB++ dataset leads to even further improvements, with performance increasing by up to 12 percentage points over the baseline (e.g., Mathstral3-8B (Mistral AI, 2024) improves from 12% to 24%). An interesting finding is that DeepSeek-Prover-v2, a whole-proof generator, showed a performance decline with LEANCOMB++, while other tree-search models performed excellently. We hypothesize that this is due to a mismatch between the model architecture and data style. ATG4CI's "learn-from-failure" mechanism generates data suited for step-by-step reasoning, which is more effectively leveraged by tree-search models. Our multi-round experiments (see Table 4) show that as training data increases, DeepSeek-Prover-v2's performance improves, suggesting it benefits from the additional data despite slower learning.

### 5.2 GENERALIZATION ANALYSIS: CROSS-DOMAIN PERFORMANCE.

We evaluated the models fine-tuned on LEANCOMB++ using two external public benchmarks: PutnamBench, CombiBench. The results (see Table 2) demonstrate that our dataset enables the models to solve a broader range of mathematical problems. All models outperform their baselines on CombiBench. Notably, the number of problems solved by InternLM2.5-Step-Prover increased

Table 1: Model Performance on **LEANCOMB-Test Set** Using Pass@1 and Pass@8 Metrics.

| Model | Pass@1 | | | Pass@8 | | |
|---|---|---|---|---|---|---|
| | Base | LEANCOMB | LEANCOMB++ | Base | LEANCOMB | LEANCOMB++ |
| *Tree Search Methods* | | | | | | |
| Mathstral3 - 8B | 9% | 15% | **19%** | 12% | 18% | **24%** |
| Llama3 - 8B | 9% | 12% | **16%** | 11% | 15% | **22%** |
| Mistral - 7B | 9% | 12% | **17%** | 13% | 18% | **23%** |
| InternLM2.5-Step | 12% | **16%** | 13% | 16% | 18% | **25%** |
| *Whole-Proof Generation* | | | | | | |
| DeepSeek-Prover-v2-7B | 19% | **21%** | 13% | 23% | **25%** | 17% |

from 2 to 7, indicating a significant improvement in its combinatorial reasoning capabilities. On PutnamBench, the fine-tuned Mathstral model even solved a previously unsolved combinatorics problem by leveraging auxiliary lemmas from LEANCOMB, highlighting the potential of our approach to enhance a model's ability to solve out-of-distribution problems.

Table 2: Model Performance on **CombiBench** and **PutnamBench** with Pass@8(%).

| Model | CombiBench | | PutnamBench | |
|---|---|---|---|---|
| | Base | LEANCOMB++ | Base | LEANCOMB++ |
| *Tree Search Methods* | | | | |
| Mathstral3 - 8B | 4/100 | **7/100** | 0/658 | **6/658** |
| Llama3 - 8B | 3/100 | **4/100** | 0/658 | **6/658** |
| Mistral - 7B | 3/100 | **7/100** | 0/658 | **6/658** |
| InternLM2.5-Step-7B | 2/100 | **7/100** | 6/658 | **7/658** |
| *Whole-proof Generation* | | | | |
| DeepSeek-Prover-v2-7B | 3/100 | **9/100** | 9/658 | **10/658** |

## 5.3 ANALYSIS AND DISCUSSION

**The Challenges of LEANCOMB Benchmark.** Although the relative performance improvements are significant, the absolute success rate on the test set highlights the challenges of the LEANCOMB benchmark. We attribute these challenges to category imbalance and inherent reasoning complexity.

Table 3: Distribution of Generated Theorems and Solved Ratio per Category.

| Theorem Category | # Generated (% of Total) | Avg. Proof Length | Solved / Total |
|---|---|---|---|
| Basic Techniques | 71003 (27.3%) | 14.83 | **18 / 34** |
| Combinatorics | 21905 (8.4%) | 34.76 | 4 / 17 |
| Calculus | 27564 (10.6%) | 25.10 | 2 / 14 |
| Probability | 33648 (13.0%) | 46.18 | 4 / 8 |
| Special Numbers | 29691 (11.4%) | 30.66 | 3 / 10 |
| Generating Functions | 12446 (4.8%) | 41.18 | 0 / 5 |
| Recurr. Rel. & Diff. | 47213 (18.1%) | 19.82 | 3 / 6 |
| Misc. & Mech. Summation | 16993 (6.5%) | 49.13 | 2 / 6 |
| **Total** | **260466 (100.0%)** | – | **39 / 100** |

Our analysis shows a strong correlation between data distribution and model performance. As seen in Table 3, the theorem generation process produces more structured proofs, such as those in the "Basic Techniques" category (27.3% of LEANCOMB++), where the success rate exceeds 50%. In

contrast, for more challenging problems like "Generating Functions" (only 4.8% of data), the success rate is 0%. The inherent complexity of the problems is a more fundamental factor. The average proof length (used as a complexity proxy) is negatively correlated with success rate. The "Basic Techniques" category, with the shortest average proof length (14.8 steps), has the highest success rate, while longer proofs, like those in "Miscellaneous Techniques" (49.1 steps) and "Generating Functions" (41.2 steps), have success rates close to zero. These findings suggest that the performance bottleneck is not just due to data imbalance, but also the lack of long-horizon reasoning capabilities, which LEANCOMB aims to rigorously test. These results highlight that LEANCOMB is a high-quality benchmark that distinguishes the "comfort zone" and "capability boundary" of models, revealing that long-horizon reasoning and complex combinatorial identities remain core challenges in ATP.

**Effect of Iterative Theorem Generation.** We analyze model performance across four rounds of iterative fine-tuning on generated theorems. As shown in Table 4, most models, including Mathstral, LLaMA3, and InternLM2.5, achieve peak performance in Round 2. The subsequent performance decline can be attributed to diversity saturation, where later rounds yield fewer novel theorems, and distribution shift, where generated data drifts from the original handcrafted distribution, potentially harming generalization. DeepSeek-Prover-v2 is an exception, reaching its highest accuracy in Round 4. Based on the overall trends, we adopt the Round 2 dataset as our final augmented training set, as it offers the best trade-off between performance and data quality across most models.

Table 4: Model performance (pass@8) across **rounds of iterative theorem generation**.

| Round | Mathstral-8B | LLaMA3-8B | Mistral-7B | InternLM2.5 | DeepSeek-v2 |
|---|---|---|---|---|---|
| Round 1 | 19% | 17% | 15% | 18% | 16% |
| Round 2 | **24%** | **22%** | **23%** | **25%** | 17% |
| Round 3 | 16% | 15% | 16% | 18% | 19% |
| Round 4 | 12% | 8% | 14% | 13% | **21%** |

**Impact of Reinforcement Learning and LLM-based Tactic Generation.** To evaluate the role of reinforcement learning and LLM-generated tactics, we conduct two ablation studies: Replace LLaMA3.1-8B tactics with naive MCTS search results; Use BFS with 100 expert-curated tactics. As shown in Table 5, both ablations produce more data than our method (BFS yields the most). However, models trained on our data perform better, surpassing MCTS and BFS by 13.5% and 14.2%, respectively, on LEANCOMB-Test set. This supports the effectiveness of reinforcement learning and LLM-guided tactic generation.

Table 5: Ablation results comparing ATG4CI with/without **RL or LLM components**.

| Method | # Candidate | #New | Success Rate(Avg.) |
|---|---|---|---|
| w/o RL (Naive MCTS) | 1,350,474 | 182,288 | 7.0% |
| w/o LLM (BFS + Expert Tactics) | 2,896,533 | 433,073 | 6.3% |
| ATG4CI (Ours) | **1,033,010** | **260,466** | **22.2%** |

# 6 CONCLUSION

We address the scarcity of formal data in combinatorics by proposing ATG4CI, based on a novel "Learn-from-Failure" paradigm that leverages unsuccessful proof attempts into Lean-verifiable theorems. By leveraging our expert-curated LEANCOMB dataset, ATG4CI generated the augmented LEANCOMB++. Experiments show that models fine-tuned on our datasets achieve significant performance gains on the in-domain test set (a 7.2% average improvement) and that their symbolic reasoning capabilities generalize to other challenging benchmarks like PutnamBench. Our work provides valuable resources for combinatorial ATP and demonstrates a scalable methodology to data creation, offering a promising template for addressing data scarcity in other mathematical domains.

ETHICS STATEMENT

This research adheres to the ICLR Code of Ethics. We explicitly acknowledge this adherence during the submission process. Our study does not involve human subjects, sensitive personal data, or any practices that would raise ethical concerns. We have made sure to comply with legal requirements, including data privacy and security, and have ensured the integrity of the research process.

We have taken the necessary steps to avoid any potential conflicts of interest or biases in the research. The methods and results are presented objectively and transparently to ensure the highest standards of research integrity.

REPRODUCIBILITY STATEMENT

We are committed to ensuring the reproducibility of our research. To this end, we provide the following resources and details.

**Code and Data Availability**: All code for our ATG4CI framework, including scripts for data generation, model fine-tuning, and evaluation, will be made publicly available. All resources, including the LEANCOMB test set, trained models, code for data generation, training, and evaluation, as well as detailed results, are available at the LEANCOMB repository.

**Experimental Details**: A comprehensive description of our experimental setup is provided in Section 5. This includes the list of all baseline models, fine-tuning hyperparameters (learning rate, batch size, optimizer, etc.), and the hardware used (NVIDIA L40 GPUs). Further details and prompt examples are available in Appendix G.

**Dataset and Evaluation**: The construction protocol, sources, and statistical properties of the LEANCOMB dataset are detailed in Section 3 and Appendix B. The evaluation protocol, including the use of Best-First Search (BFS) and the pass@k metrics, is described in Section 5. All proofs were verified using Lean 4 (v4.14.0) and its corresponding Mathlib4 library.

LLM USAGE STATEMENT

A Large Language Model (LLM) was used to assist in the writing and polishing of this manuscript. Specifically, an LLM aided in improving grammar, clarity, and readability through tasks such as sentence rephrasing and proofreading.

It is critical to note that the LLM was not involved in any core scientific aspects of this work, including ideation, formulation of the research methodology, experimental design, or data analysis. All research concepts, ideas, and scientific conclusions were developed exclusively by the human authors. The LLM's contribution was strictly limited to improving the linguistic quality of the text. The authors take full responsibility for all content in this manuscript, including the accuracy and integrity of any text modified by the LLM.

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

## A  AN ILLUSTRATIVE EXAMPLE FOR ATG4CI

We now illustrate the end-to-end workflow of ATG4CI using a single identity, $sum\_mul\_congr$, from our seed dataset.

**Example 1.** Consider a combinatorial identity and its formalization in Lean 4:

$$\sum_{k=1}^{n} nC_{n-1}^{k-1} = n\sum_{l=0}^{n-1} C_{n-1}^{l}. \tag{1}$$

```
theorem sum_mul_congr {n : ℕ}:
  Σ k ∈ Finset.Ico 1 (n + 1), n * Nat.choose (n - 1) (k - 1) =
  n * Σ l ∈ Finset.range n, Nat.choose (n - 1) l := by
  rw [Finset.mul_sum]
  rw [Finset.sum_Ico_eq_sum_range]
  simp only [add_tsub_cancel_right,
    add_tsub_cancel_left]
```

Listing 1: The root theorem 'sum_mul_congr' in Lean 4.

We formalize identity ( 1) as a root theorem named $sum\_mul\_congr$. Its proof concisely demonstrates typical multi-step reasoning in Lean 4: first, the common factor $n$ is factored out of the summation using the $rw\,[Finset.mul\_sum]$ tactic; next, the $rw\,[Finset.sum\_Ico\_eq\_sum\_range]$ tactic is used to unify the summation bounds; and finally, the $simp$ tactic is called to complete the algebraic simplification, reaching the "$no\;goals$" state, which signifies the proof is complete. This example clearly reflects the precision and logical steps required for proving the identities within LEANCOMB.

**Stage 1: *P3s* (Partial Proof Paths) Construction**

During the *P3s* construction process, the root theorem $sum\_mul\_congr$ shown in Example 1 transitions through three sequential tactics and eventually reaches the "no goals" state at the leaf node, resulting in a four-layer proof tree. From this tree, two *P3s* are extracted: one from the root to the state after applying "$rw\,[Finset.mul\_sum]$", and another from the root to the state after applying "$rw\,[Finset.sum\_Ico\_eq\_sum\_range]$":

```
theorem P3s_1 {n : ℕ}:
  Σ k ∈ Finset.Ico 1 (n + 1), n * Nat.choose (n - 1) (k - 1) =
  n * Σ l ∈ Finset.range n, Nat.choose (n - 1) l := by
  rw [Finset.mul_sum]
```

Listing 2: The first *P3s* of the root theorem.

```
theorem P3s_2 {n : ℕ}:
  Σ k ∈ Finset.Ico 1 (n + 1), n * Nat.choose (n - 1) (k - 1) =
  n * Σ l ∈ Finset.range n, Nat.choose (n - 1) l := by
  rw [Finset.mul_sum]
  rw [Finset.sum_Ico_eq_sum_range]
```

Listing 3: The second *P3s* of the root theorem.

**Stage 2: Candidate Theorem Generation (Learning from Failure)**

The *P3s* is then fed to an LLM, which attempts to continue the proof by generating new candidate tactics. In many explorations, the model-generated path does not immediately complete the proof. The code below shows one such failed candidate proof-path, where the predicted tactics (Lines 7–10) do not close the goal:

```
1  theorem cp_1 {n : ℕ} :
2   Σ k ∈ Finset.Ico 1 (n + 1), n * Nat.choose (n - 1) (k - 1) =
3   n * Σ k ∈ Finset.range n, Nat.choose (n - 1) k := by
4    rw [Finset.mul_sum]
5    -- Prediction starts here!
6    rw [range_eq_Ico]
7    rw [sum_Ico_eq_sum_range]
8    rw [add_tsub_cancel_right]
9    rw [range_eq_Ico]
10   -- Prediction ends, goal is not closed
```

Listing 4: A candidate proof-path that fails to close the goal.

When this path terminates, the final unproven subgoal is identified. Our framework's core innovation is to transform this "failure" into a new, valid theorem. The original theorem $sum\_mul\_congr$ is repurposed as a new hypothesis $h$, and the final subgoal from the failed path becomes the new conclusion:

$$\sum_{k=0}^{n} nC_{n-1}^{1+k-1} = \sum_{l=0}^{n} nC_{n-1}^{l}. \tag{2}$$

The proof for this new theorem is constructed by replaying the tactics from the failed path on the hypothesis and using the assumption tactic to finalize it, as shown in the new candidate theorem CT_1:

```
1  theorem CT_1 (n : ℕ)
2    (h : Σ k ∈ Finset.Ico 1 (n + 1), n * Nat.choose (n - 1) (k - 1)
3      = n * Σ l ∈ Finset.range n, Nat.choose (n - 1) l) :
4    -- Candidate theorem
5    Σ k in range n, n * Nat.choose (n - 1) k =
6    Σ x in Ico 0 n, n * Nat.choose (n - 1) x := by
7      rw [mul_sum] at h
8      rw [range_eq_Ico] at h
9      rw [sum_Ico_eq_sum_range] at h
10     rw [add_tsub_cancel_right] at h
11     assumption
```

Listing 5: A new candidate theorem generated from the failed path.

As shown in Lines 3–5, the root theorem is adopted as the new hypothesis $h$. Then, in Lines 10–13, all tactics along the candidate path are sequentially applied to rewrite $h$ until it aligns with the target goal. Finally, the tactic "assumption" in Line 14 is applied to conclude the proof.

**Stage 3: Theorem Validation**

The newly generated candidate theorem, like CT_1, then enters the final validation stage. Its proof is first formally verified by Lean. If successful, the theorem is deduplicated to ensure it is unique from other generated theorems. If the theorem contained errors, it would be sent to an automated correction module. Only correct and unique theorems that pass all checks are added to the LEANCOMB++ dataset.

## B    LEANCOMB DATASET

The LEANCOMB dataset is constructed based on authoritative literature in the field of classical combinatorics (Spivey, 2019; Gould, 1972; Shi, 2001), aiming to provide a high-quality mathematical reasoning benchmark to support automated theorem proving for combinatorial identities. The dataset comprises a total of 727 theorems, with the training set consisting of 627 theorems (including 418 combinatorial identities and 209 lemmas), while the test set contains 100 identities.

When annotating, we strive to cover more combinatorial mathematics fields. The specific classification of theorems in LEANCOMB is shown in Table 6. All identities in the training set and test sets are

| | Theorem Categories | Training Set | Test Set |
|---|---|---|---|
| | The Generalized Binomial Coefficient | 42 | 16 |
| Basic Techniques | Absorption Identity | 35 | 12 |
| | Binomial Inversion | 46 | 6 |
| | Choosing with Replacement | 26 | 6 |
| Combinatorics | Alternating Binomial Sums & Involutions | 23 | 6 |
| | Inclusion-Exclusion | 27 | 5 |
| | Differentiation | 20 | 5 |
| Calculus | Integration | 26 | 3 |
| | Beta Integral & Gamma Function | 22 | 6 |
| Probability | Binomial & Hypergeometric Distributions | 13 | 5 |
| | Expected Values & Moments | 16 | 3 |
| Special Numbers | | 25 | 10 |
| Generating Functions | | 29 | 5 |
| Recurrence Relations & Finite Differences | | 56 | 6 |
| Miscellaneous Techniques & Mechanical Summation | | 12 | 6 |
| Total | | 418 | 100 |

Table 6: Theorem categories with training and test set counts.

classified into 8 categories, including Combinatorics, Calculus, Probability, and others. The Basic Techniques category contains the most significant samples in training and test sets, with 123 and 40 samples, respectively. In the "Basic Techniques" category, the Binomial Coefficient and Binomial Inversion, along with the Recurrence Relations & Differences categories, have relatively higher sample counts, totaling 42, 46, and 56, respectively. Additionally, we have formally introduced 9 new mathematical definitions, covering fundamental concepts such as Bell numbers, Stirling numbers of the first and second kinds, and combinations and permutations, thereby enhancing the expressiveness of the benchmark dataset. All resources, including the LEANCOMB test set, trained models, code for data generation, training, and evaluation, as well as detailed results, are available at the LEANCOMB repository.

This dataset encompasses numerous classical theorems in combinatorics, including Negative Binomial Series, Binet's Formula, Trinomial Revision, and Cassini's Identity. For example:

$$\text{Negative Binomial Series:} \frac{1}{(1-x)^{n+1}} = \sum_{k=0}^{\infty} \binom{n+k}{n} x^k,$$

$$\text{Binet's Formula:} \quad F_n = \frac{\phi^n - \psi^n}{\sqrt{5}},$$

$$\text{Cassini's Identity:} \quad F_{n+1}F_{n-1} - F_n^2 = (-1)^n.$$

To systematically illustrate the diversity of the dataset and the core mathematical structures involved, we have selected several representative combinatorial identity examples and categorized them based on their roles within the dataset. The selected examples are divided into two parts: one from the training set and the other from the test set. Each part includes various types of combinatorial identities, covering key topics such as the properties of Stirling numbers, classical combinatorial identities, and their variants. These examples not only provide an intuitive understanding of the dataset's composition but also establish a solid foundation for subsequent theoretical analysis and model evaluation.

## B.1 Training Set

**idt_15:**

Goal: $\sum_{k=0}^{m} \binom{n+k}{k} = \binom{n+m+1}{m}$

**idt_101**

Goal: $\sum_{k=0}^{n} \binom{n}{k} \frac{(x_2^{k+1} - x_1^{k+1})y^{n-k}}{(k+1)^2} = \frac{1}{n+1} \sum_{k=0}^{n} \frac{((x_2+y)^{k+1} - (x_1+y)^{k+1})y^{n-k}}{k+1}$

**idt_105**

Premises: $a, b \in \mathbb{R}_{\geq 0}$

Goal: $B(a, b) = \frac{\Gamma(a)\Gamma(b)}{\Gamma(a+b)}$

**idt_106**

Premises: $-1 < n - k$

Goal: $\frac{k!}{n^{\underline{k}}} = (n+1) \int_0^1 x^k (1-x)^{n-k} dx$

**idt_109**

Premises: $0 < x$

Goal: $\sum_{k=0}^{n} \binom{n}{k} \frac{(-1)^k}{k+x} = \frac{n!}{x(x+1)\dots(x+n)}$

**idt_112**

Goal: $\sum_{k=0}^{n} \binom{n}{k} k(k+m) = n(n+2m+1)2^{n-2}$

**idt_115**

Goal: $\sum_{k=0}^{r} \binom{n}{r-k} \binom{m+k}{m} (-1)^k = \binom{n-m-1}{r}$

**idt_117**

Goal: $\sum_{k=0}^{n} \binom{n}{k} \frac{(n+1)(n+2)(n+3)}{(k+1)(k+2)(k+3)} = 2^{n+3} - 1 - (n+3) - \frac{(n+2)(n+3)}{2}$

**idt_121**

Goal: $\sum_{n=0}^{\infty} \frac{2^n}{(2n+1)\binom{2n}{n}} = \frac{\pi}{2}$

**idt_124**

Goal: $\frac{1}{\binom{n}{k}} = 2(n+1) \int_0^{\frac{\pi}{2}} \sin^{2k+1}\theta \cos^{2n-2k+1}\theta d\theta$

**idt_131**

Premises: $0 \leq p \leq 1$

Goal: $\sum_{k=0}^{n} \binom{n}{k} p^k (1-p)^{n-k} k^2 = n^2 p^2 + np(1-p)$

**idt_132**

Goal: $\sum_{k=0}^{m} \binom{m}{k} \binom{n}{r-k} k = \binom{m+n}{r} \frac{rm}{m+n}$

**idt_179**

Premises: $n \leq m$

Goal: $\Delta^n F_m = F_{m-n}$

**idt_212**

Goal: $\sum_{k=1}^{n} \binom{n}{k} \frac{(-1)^k}{k(k+1)\cdots(k+m)} = -\frac{H_{n+m} - H_m}{m!}$

**idt_285**

Premises: $0 < y$

Goal: $\sum_{k=0}^{n} \binom{n}{k} i^k y^{n-k} = (y^2 + 1)^{n/2} e^{i\,n\,\arctan(1/y)}$

**idt_318**

Goal: $\sum_{k \geq 0} \binom{n}{2k+1} 3^k (-1)^k = \frac{2^n}{\sqrt{3}} \sin \frac{n\pi}{3}$

## B.2    TEST SET

**test_002**

Goal: $n^{\overline{k+1}} = (k+1)! \times \binom{n+k}{k+1}$

**test_005**

Premises: $1 \leq n$

Goal: $\sum_{k=1}^{n} \frac{1}{k} \binom{2(k-1)}{k-1} \binom{2(n-k)}{n-k} = \frac{1}{2} \binom{2n}{n}$

**test_011**

Premises: $2n \leq m; 1 \leq n$

Goal: $\sum_{k=0}^{n} (-1)^k \binom{n}{k} \binom{m-2k}{n-1} = 0.$

**test_025**

Goal: $\sum_{k=0}^{\lfloor n/2 \rfloor} \binom{n-k}{k} = \frac{1}{\sqrt{5}} \left( \left( \frac{1+\sqrt{5}}{2} \right)^{n+1} - \left( \frac{1-\sqrt{5}}{2} \right)^{n+1} \right).$

**test_037**

Goal: $\sum_{k=0}^{n} (-1)^k \binom{n}{k} \binom{m-k}{r} = \binom{m-n}{r-n}.$

**test_054**

Premises: $1 \leq n$

Goal: $\sum_{k=0}^{m} \binom{k+n-1}{n-1} = \binom{n+m}{n}.$

**test_069**

Premises: $3 \leq n$

Goal: $n^3 = 6 \binom{n}{3} + 6 \binom{n}{2} + \binom{n}{1}$

**test_076**

Goal: $\sum_{k=0}^{n-1} (-1)^k \left( \cos\left(\frac{k\pi}{n}\right) \right)^n = \frac{n}{2^{n-1}}.$

**test_088**

Premises: $1 \leq n$

Goal: $\sum_{k=0}^{n} \binom{2n}{k} = 2^{2n-1} + \frac{1}{2} \binom{2n}{n}$

**test_100**

Premises: $k \leq n$

Goal: $\binom{n}{k} \cdot \text{Beta}(k+1, n-k+1) = \frac{1}{n+1}$

## B.3    DATA CONTAMINATION CHECK

We examine potential data contamination between the LEANCOMB dataset and established automated theorem proving (ATP) datasets and benchmarks. Specifically, we check for overlaps with Mathlib4, as well as with benchmark suites including MiniF2F, PutnamBench, ProofNet, FIMO, and CombiBench.

During this process, we identified 12 combinatorial identities in the training set that overlap with the mathlib corpus, and one test problem that overlaps with CombiBench. To prevent data leakage, we removed all overlapping examples from the training set, while annotating their original paths in Mathlib. And, the overlapping test case was also removed.

## C  MORE DETAILS ABOUT ATG4CI

### C.1  AN ILLUSTRATIVE EXAMPLE FOR ATG4CI

For the LEANCOMB dataset, data augmentation is performed using our theorem generator, ATG4CI.
Some of the theorems in the dataset are augmented to generate up to 4,000 new instances in one
iteration, resulting in a total of 260,466 newly generated theorems. Among these, 56,747 erroneous
theorems are successfully corrected. Below, we demonstrate the workflow of ATG4CI using a typical
example, which corresponds to the mathematical formula $\sum_{k=0}^{n} \binom{n}{k} F_{m+k} = F_{2n+m}$:

```
1  import Mathlib
2  import Theorem.valid.idt_179
3
4  open Finset Nat
5
6  theorem idt_182 (m n : ℕ) : Σ k in Finset.range (n + 1),
       Nat.choose n k * Nat.fib (m + k) = Nat.fib (2 * n + m) := by
7    suffices Σ k in Finset.range (n + 1), Nat.choose n k * Nat.fib
       (m + k) = (Nat.fib (2 * n + m) : ℝ) by
8      norm_cast at this
9    let g := fun k => Nat.fib (m + n + k)
10   -- $sum_{k=0}^{n} \binom{n}{k} * fib(m+k) = \sum_{k=0}^{n}
       \binom{n}{k} * fib(m+n-n+k)$
11   have h₁: Σ k in Finset.range (n + 1), Nat.choose n k * Nat.fib
       (m + k) =
12     Σ k in Finset.range (n + 1), Nat.choose n k * Nat.fib (m + n
       - n + k) := by
13     congr! 1 with k hk
14     simp at hk
15     congr; omega
16   rw [h₁]
17   -- $\sum_{k=0}^{n} \binom{n}{k} * fib(m+n-n+k) = \sum_{k=0}^{n}
       \binom{n}{k} * fib(m+n-k)$
18   rw [← Finset.sum_flip]
19   have h₂: Σ k ∈ range (n + 1), n.choose (n - k) * fib (m + n - n +
       (n - k)) = Σ k ∈ range (n + 1), n.choose k * fib (m + n - k)
       := by
20     congr! 1 with k hk
21     simp at hk
22     rw [choose_symm (by linarith)]
23     congr; omega
24   rw [h₂]
25   -- $\sum_{k=0}^{n} \binom{n}{k} * \text{fib}(m+n-k) =
       \sum_{k=0}^{n} \binom{n}{k} * \sum_{j=0}^{k} \binom{k}{j} *
       g(j)*(-1)^k*(-1)^j$
26
27   ... ...
28
29   have h₇: Σ k ∈ range (n + 1), n.choose k * g k * (-1) ^ k * Σ x
       ∈ range (n - k + 1), (n - k).choose (x) * (-1 : ℝ) ^ (x + k) =
30     Σ k ∈ range (n + 1), n.choose k * g k * Σ x ∈ range (n - k +
       1), (n - k).choose (x) * (-1 : ℝ) ^ x := by
31     congr! 1 with k _
32     rw [mul_assoc, mul_sum]
33     congr 1
34     congr! 1 with j _
35     rw [mul_comm, pow_add, ← mul_assoc, mul_assoc, ← pow_add, ←
       two_mul, pow_mul]
36     ring
37   rw [h₇, ← Finset.sum_range_add_sum_Ico _ (m := n) (by omega),
       show 2 * n + m = m + n + n by omega]
38   simp [g]
39   apply sum_eq_zero
```

```
40    intro k hk
41    rw [_root_.mul_eq_zero]
42    right
43    simp at hk
44    rw [show (0 : ℝ) = (-1 + 1) ^ (n - k) by simp; rw [zero_pow (by
        omega)], add_pow]
45    simp [mul_comm]
```

Combinatorial Identity 182 encompasses key concepts such as binomial coefficients, Fibonacci numbers, and generating functions. The proof consists of 77 lines involving 23 distinct tactics, resulting in a proof length and tactic diversity of 23. Structurally, it forms a 24-layer proof tree, from which 22 distinct *P3s* can be extracted. The longest and shortest *P3s* are highlighted as follows:

```
1  theorem P3s_1 (m n : ℕ) : Σ k in Finset.range (n + 1), Nat.choose
      n k * Nat.fib (m + k) = Nat.fib (2 * n + m) := by
2    suffices Σ k in Finset.range (n + 1), Nat.choose n k * Nat.fib
      (m + k) = (Nat.fib (2 * n + m) : ℝ) by
3      norm_cast at this
4    ... ...
5    rw [h₁]
```

```
1  theorem P3s_22 (m n : ℕ) : Σ k in Finset.range (n + 1),
      Nat.choose n k * Nat.fib (m + k) = Nat.fib (2 * n + m) := by
2    suffices Σ k in Finset.range (n + 1), Nat.choose n k * Nat.fib
      (m + k) = (Nat.fib (2 * n + m) : ℝ) by
3      norm_cast at this
4    ... ...
5    rw [show (0 : ℝ) = (-1 + 1) ^ (n - k) by simp; rw [zero_pow (by
      omega)], add_pow]
```

Based on these *P3s*, our RLSCI framework facilitates tactic prediction, enabling the generation of candidate theorems. We further construct their proofs, followed by deduplication and correctness verification. Below, we present a selection of intriguing combinatorial identities generated by our theorem generator using the aforementioned examples:

**idt_182_3_12:**

$$\sum_{k=0}^{n} \binom{n}{k} \sum_{j=0}^{k} \binom{k}{j} F_{m+n+j}(-1)^j(-1)^k = F_{2n+m}$$

**idt_182_1_3:**

$$\sum_{k=0}^{n} \binom{n}{k} F_{m+n-k} = F_{2n+m}$$

**idt_182_0_5:**

$$\sum_{x=0}^{n}\sum_{k=0}^{x} \binom{n}{k}\binom{n-k}{x-k} F_{m+n+k}(-1)^{x+k} = F_{2n+m}$$

**idt_182_0_72**

$$\sum_{k=0}^{n} \binom{n}{k} F_{m+n-k} = F_{2n+m}$$

**idt_182_6_5**

$$\sum_{k=0}^{n} \binom{n}{k} F_{m+n+k}(-1)^k \sum_{x=0}^{n-k} \binom{n-k}{x}(-1)^{x+k} = F_{2n+m}$$

**idt_182_6_23**

$$\sum_{k=0}^{n-1} \binom{n}{k} F_{m+n+k} \sum_{j=0}^{n-k} \binom{n-k}{j} (-1)^j = 0$$

**idt_182_6_49**

$$\binom{n}{k} F_{m+n+k} \sum_{x=0}^{n-k} \binom{n-k}{x} (-1)^x = 0$$

**idt_182_2_52**

$$\sum_{x=0}^{n-k} \binom{n-k}{x} (-1)^x = 0$$

**idt_182_2_45**

$$\sum_{r=0}^{n} \binom{n}{n-r} F_{m+(n-r)} = F_{2n+m}$$

## C.2 DATA CONTAMINATION CHECK

To enable a more comprehensive evaluation, we further examined overlaps between the LEAN-COMB++ dataset and existing formalized resources, including Mathlib and the Lean Community Workbook. To prevent data contamination, we deduplicated at the tactic-level by converting each proof into a sequence of (state, tactic) pairs and removing any duplicated tactic sequences across datasets.

## C.3 THEOREM CORRECTION DETAILS

**Theorem Correction**: After deduplication, as some candidate theorems may still contain expression errors or fail to pass the proof process, correctness refinement is performed. After candidate theorems are verified successfully by interacting with Lean, they are directly added to the generated dataset $G^*$. Those that fail verification are categorized by error type and corrected using the corresponding correction methods. The corrected theorems are then added to $G^*$. The following section elaborates on the error types and their corresponding correction tactics.

- *Incomplete Error:* An incomplete error occurs when a candidate tactic generates multiple subgoals, at least one subgoal remains unproven, preventing the completion of the theorem's proof. The standard MCTS (Coulom, 2006) method can be applied to recover incomplete theorems and generate additional proof steps to complete the proof.

- *Type Errors:* Type errors arise due to Lean's representation methods, where variable types in subgoals are not always explicitly stated during interactions with Lean or during the data extraction process. To address this, we identify the type information from the original theorem and annotate the generated theorems with correct type labels.

- *Logical Errors:* Logical errors occur when applying a tactic that generates multiple subgoals, but at least one contains a logical inconsistency, preventing further progress in the proof. Theorems with such logical inconsistencies are considered irreparable and must be discarded.

During the theorem generation process, we classify theorems and verify the correctness of candidate theorems. Incorrect theorems are grouped separately, and different correction methods are applied based on their types.

The standard MCTS with our fine-tuned Llama 3.1 corrects these theorems. We set the number of candidate tactics (i.e., visit counts) per node to 16 and the number of simulations per node to 100. It generates full-proof search trees through selection, expansion, and backpropagation, and complete-proof steps are engendered accordingly. The selection phase considers the average reward and exploration of nodes, using the UCB1 algorithm to select the optimal node (Auer et al., 2002):

| Error Types | Error Example | Correction Example |
|---|---|---|
| Incomplete Proofs | $ErrorExample_1(n : \mathbb{N})$
(goal : the goal of sum_mul_congr) :
$\sum_{k=0}^{n} kC_{2n+1} = 2n \sum_{k=0}^{n} C_{2n}^{k+1} \sum_{k=0}^{n} C_{2n}^{k}$ := by
$rw[range\_eq\_Ico]$ at goal
assumption | $CorrectedTheorem_1(n : \mathbb{N})$
(goal : the goal of sum_mul_congr) :
$\sum_{k=0}^{n} kC_{2n+1} = 2n \sum_{k=0}^{n} C_{2n}^{k} + 1 \sum_{k=0}^{n} C_{2n}^{k}$ := by
$rw[range\_eq\_Ico]$ at goal
$rw[add\_mul]$ at goal
assumption |
| Type Errors | $ErrorExample_2(m : \mathbb{N})(hm : 0 < m) :$
$2 + {\color{red}-1}/(m+1) = (2m+1)/(m+1)$ | $Corrected\_Theorem_2(m : \mathbb{N})(hm : 0 < m) :$
$2 + (-1 : R)/(m+1) = (2m+1)/(m+1)$ |
| Redundant Steps | $ErrorExample_3(n : \mathbb{N}) :$
$2n + 1 - n = n + 1$ := by
$rw[two\_mul]$
$rw[add\_assoc]$
$rw[add\_comm]$
simp
${\color{red}rw[two\_mul]}$ | $CorrectedTheorem_3(n : \mathbb{N}) :$
$2n + 1 - n = n + 1$ := by
$rw[two\_mul]$
$rw[add\_assoc]$
$rw[add\_comm]$
simp |

Table 7: The Error Types and Correction Process

$$UCB_1 = \frac{W_i}{N_i} + C \times \sqrt{\frac{lnN_p}{N_i}}. \tag{3}$$

Detailed descriptions of the different types of errors and their respective correction methods are provided in the table 7.

**Example 1: Incomplete Proofs**

```
theorem congr_Ico_succ__2__73(n : ℕ) :
  Σ k in Ico 1 (n + 1), k * Nat.choose (n - 1) (k - 1) = Σ l in
  Ico 0 n, (l + 1) * Nat.choose (n - 1) l := by
  rw [sum_Ico_eq_sum_range]
  simp
  refine' sum_congr rfl fun y _ => _
  rw [add_mul]
  rW [choose_eq_zero_of_lt]
  rw [add_comm]
```

Listing 6: Type Error: unsolved goal $n - 1 < y$.

**After Corrected:**

```
theorem congr_Ico_succ_2_73 (n : ℕ) :
  Σ k in Ico 1 (n + 1), k * Nat.choose (n - 1) (k - 1) = Σ l
  in Ico 0 n, (l + 1) * Nat.choose (n - 1) l := by
  rw[sum_Ico_eq_sum_range]
  simp
  refine' sum_congr rfl fun x _ => _
  simp
  rw [add_comm]
  exact Or.inl rfl
```

**Example 2: Type Errors**

```
theorem sum_mul_add_distrib__0__36(n : ℕ)(h : Σ k in range (n +
  1), (k + 1) * Nat.choose n k = Σ k in range (n + 1), ( k *
  Nat.choose n k + 1 * Nat.choose n k)) :
```

```
2  (1 + y) * Nat.choose n y = y * Nat.choose n y + Nat.choose n y :=
      by
3  refine' sum_congr rfl fun y _ => _
4  simp [mul_assoc] at h
5  rw [add_comm] at h
6  assumption
```

Listing 7: Type Error : metavariables  AddCommMonoid  ?m.90801.

**After Corrected:**

```
1  theorem sum_mul_add_distrib__0__36(n : ℕ)(h : Σ k in range (n +
      1), (k + 1) * Nat.choose n k = Σ k in range (n + 1), (k *
      Nat.choose n k + 1 * Nat.choose n k)) :
2  (1 + y) * Nat.choose n y = y * Nat.choose n y + Nat.choose n y
      := by
3  rw [add_mul]
4  rw [one_mul]
5  rw [add_comm]
```

# D  LEAN4KIT

In our automated theorem generator ATG4CI, Lean4Kit plays a key role in data extraction and interaction. During the experimental evaluation and testing phase, Lean4Kit assists LLMs in theorem proving through interaction with Lean 4. In fact, given any repo in Lean 4, our offline toolkit can convert Lean files into JSON format data, extracting all state-tactic pairs.

## D.1  DATA EXTRACTION FROM LEAN CODES

The complete Lean code (including imports) is converted into a JSON-formatted data structure tree (infotree) in the static extraction process. The conversion function *run_all_tactics(self, code, env=None, verbose=True)* returns data in a format that includes the tactic, as well as the before-and-after states of the goals (goalsBefore and goalsAfter), for example:

**pp**: rw [abelidentity_eq_add]
**name** : Lean.Parser.Tactic.rwSeq
**goalsBefore :**

```
1  n : ℕ
2  x y : ℝ
3  hn1 : 1 ≤ n
4  hx : x ≠ 0
5  hy : y ≠ 0
6  ⊢ abelidentity x y (−1) (−1) n =
7  (1 / x + 1 / y) * (x + y + ↑n) ^ (n − 1)
```

**pp**: rw [abelidentity_eq_add]
**name** : Lean.Parser.Tactic.rwSeq

**goalsAfter :**

```
1  n : ℕ
2  x y : ℝ
3  hn1 : 1 ≤ n
4  hx : x ≠ 0
5  hy : y ≠ 0
6  ⊢ abelidentity x (y + 1) (−1) (−1 + 1) (n − 1) +
       abelidentity (x + 1) y (−1 + 1) (−1) (n − 1) = (1 / x +
       1 / y) * (x + y + ↑n) ^ (n − 1)
7
8  case hn
9  n : ℕ
10 x y : ℝ
11 hn1 : 1 ≤ n
12 hx : x ≠ 0
13 hy : y ≠ 0
14 ⊢ n ≥ 1
```

## D.2 DYNAMIC INTERACTION WITH LEAN FOR THEOREM PROVING

In the dynamic interaction process, we interact with Lean to perform automated theorem proving. The process mainly relies on the following functions:

- *run_import*(self, code, env=None, verbose=False) : Used to import necessary environments and dependencies.

- *new_thm*(self, code, env=None, verbose=False) : Generates a new theorem based on the provided theorem description.

The provided code parameter represents the description of the initial theorem, for example:

```
1  theorem idt_84 (n : ℕ)(h : m < n): Σ k in range (n + 1), n.choose
       k * (n − k) ^ m * (−1 : ℝ) ^ k = 0 := by sorry
```

This function returns an initial state for subsequent tactic applications:

- *run_tactic*(self, tactic, proofState, cmd_type= 'tactic', verbose=False): Executes a tactic and returns the new state after the tactic is applied.

- *run_have_tactic*(self, tactic, proofState, cmd_type='have', verbose=False) : Executes a "have" tactic.

During the interaction, we can also use the function $is\_correct\_and\_finished(self, code, verbose = False, timeout = 160)$ to check if the theorem is correct and whether the proof is complete, with the judgment based on the information view (Lean Infoview) on the right side.

> **ProcessingCmd :**
>
> ```
> 1 example (a b c : 2115)(h : a = b): a ^ 2 + c= b ^ 2 + c:= by
>     sorry""
> 2 env: 0
> ```
>
> **proofstates :** [0],
> **goals :**
>
> ```
> 1  a b c : N  h : a = b ⊢ a ^ 2 + c = b ^ 2 + c
> ```
>
> **error**: False
> **messages**: [{'severity': 'warning', 'pos': {'line': 1,'column': 0}, 'endPos': {'line': 1,'column': 7}, 'data': "declaration uses 'sorry'"}]
> **sorries**: [{'proofState' : 0, 'pos': {'line' : 1,'column' : 58}, 'goals', 'endPos' : {'line': 1,'column': 63}}]
> **is finish**: False

The returned Tactic State format includes the following key fields:

- *messages*: Contains information during the interaction, such as "no goals," "declare use sorry," "unknown tactic," etc.

- *proofstates* : An integer list uniquely identifies the current state, where each integer corresponds to a subgoal's state.

- *goals* : The current subgoals.

- *error*, *finishFlag* : These parameters are assigned by analyzing messages . The error indicates whether an error occurred (True for error), and finishFlag indicates whether the proof is complete (True for completed proof).

# E   MORE STATISTIC RESULTS

## E.1   PREDICTION STEPS DISTRIBUTION IN LEANCOMB++ DATASET.

Fig. 6 presents the distribution of theorem counts across different prediction steps, categorized into four groups: deduplicated, correct, corrected, and new theorems. The data reveal a peak around a prediction step of 6, indicating that most theorems are concentrated at this length regardless of their classification. The number of deduplicated theorems exhibits the highest count, peaking at $121,457$. In contrast, the correct and new theorems follow similar trends but at lower magnitudes, suggesting that a significant proportion of generated theorems are either duplicates or require correction. The corrected theorems, represented by the green triangles, remain consistently lower than the other categories, highlighting the challenges in refining generated theorems through correction mechanisms.

Additionally, the distribution demonstrates a sharp decline in theorem numbers beyond a prediction step of 6, implying that longer proof sequences are less frequent and potentially more challenging to generate and verify. These findings suggest that optimizing theorem generation should focus on mid-range prediction steps, where the balance between uniqueness, correctness, and novelty is most favorable.

## E.2   THEOREM GENERATION STATISTICS ACROSS FOUR ITERATIVE ROUNDS

Figure 7 presents the statistics of theorem generation across four iterative rounds. In each round, a large number of candidate theorems were initially generated by ATG4CI, followed by a deduplication process to eliminate redundancies. As shown, the number of candidate theorems gradually decreased from 592.81K in Round 1 to 393.87K in Round 4, indicating a convergence trend in the generation process. After deduplication, approximately $66\%$ of the candidate theorems were retained, suggesting that a significant portion of redundancy existed among the initial outputs.

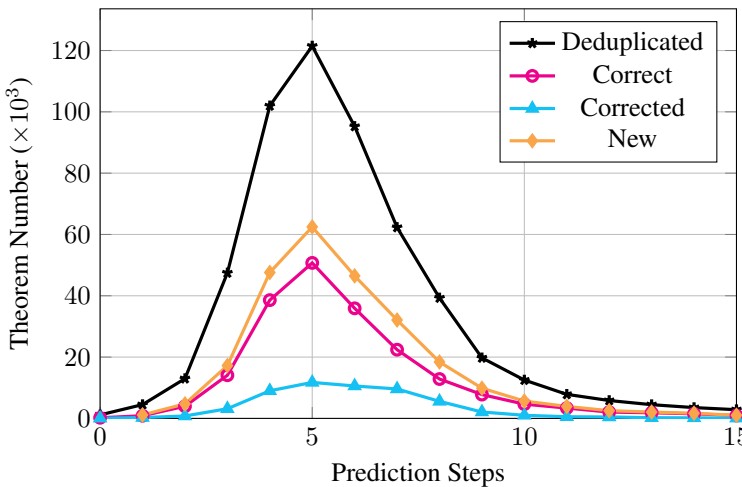

Figure 6: Distribution of Theorem Numbers on Prediction Steps

Subsequent correctness verification further reduced the number of theorems, with only about 17–24% of the deduplicated theorems passing the verification in each round. Moreover, a portion of the incorrect theorems were successfully repaired, yielding additional correct theorems as indicated by the yellow bars. Notably, the number of corrected theorems was consistently substantial, particularly in Round 2, where 160.39k theorems were recovered.

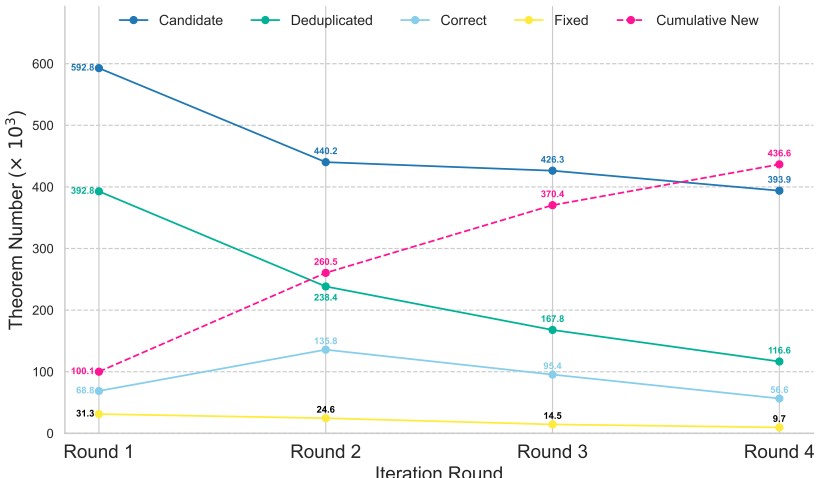

Figure 7: Theorem generation statistics across iterations. **Candidate** denotes the number of candidate theorems generated by ATG4CI, **Deduplicated** are the theorems retained after deduplication, **Correct** are those that passed correctness verification, **Fixed** are the repaired incorrect theorems, and **Cumulative New** represents the cumulative count of new theorems.

In total, each round produced 68.77K, 135.80K, 95.38K, and 56.57K correct theorems, respectively, reflecting both the effectiveness of the repair mechanism and the increasing difficulty of generating novel valid theorems in later rounds. Overall, these results demonstrate that while the initial generation process produces a large volume of candidates, post-processing steps such as deduplication, verification, and correction are critical to ensuring the quality and validity of the final theorem set.

### E.3 DISTRIBUTION OF THEOREMS IN $E^*$ BY THEOREM TYPE COUNT

Fig. 8 shows the distribution of deduplicated, correct, corrected, and new theorems by proof steps in the enhanced dataset $E^*$, including newly added data. Deduplicated theorems peak at proof step 6

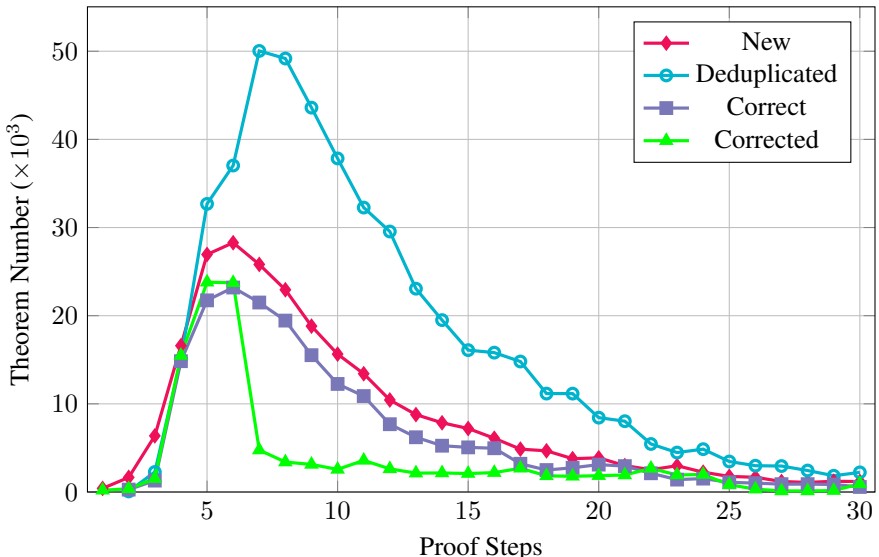

Figure 8: Theorem distribution by proof Steps

with approximately $50,000$ and decline as steps increase, indicating shorter proofs dominate. Correct theorems peak slightly earlier at step 5 with $22,000$, showing that many deduplicated theorems fail correctness checks. Corrected theorems are fewer and decrease rapidly after step 5, suggesting that errors in longer proofs are harder to address. The new data aligns with these trends, peaking at step 6 with $28,000$ theorems but maintaining a steady count for longer proofs, highlighting improved generation of extended proofs yet persistent challenges in verification.

These results emphasize two key challenges in automated theorem proving: the gap between deduplicated and correct theorems, particularly for longer proofs, underscores the need for more robust verification methods, and the rapid decline in corrected theorems highlights the difficulty of resolving errors in complex proofs. Improving proof generation and error correction tactics—especially for longer proofs—remains a crucial direction, alongside integrating advanced validation mechanisms to enhance correctness and diversity.

## F  MORE EXPERIMENTAL RESULTS

### F.1  EFFECT OF CANDIDATE TACTIC QUANTITY ON THEOREM GENERATION

We first investigate how the number of candidate tactics affects theorem generation quality. Table 8 summarizes the results of ATG4CI under different settings (4, 8, and 16 candidates).

In the first iteration, setting 16 candidate tactics produced 592,811 theorems, which reduced to 392,818 after deduplication. Among them, 68,771 theorems were verified as correct, and 31,306 erroneous theorems were successfully corrected, resulting in a total of 100,077 new theorems. In comparison, setting 4 and 8 candidate tactics resulted in 22,691 and 74,136 new theorems, respectively. In the second iteration, the proportion of correct theorems significantly increased, reaching up to 75.0%, and the new theorem generation rates also improved from 10.7%, 20.3%, and 16.9% to 33.7%, 41.7%, and 36.4%, respectively. While the 16-tactic setting produced the largest number of theorems, the 8-tactic setting achieved the best success rate for generating new theorems.

### F.2  AUTOMATED PROOF RESULTS OF OUR MODELS ON LEANCOMB-TEST

In this section, we analyze the proof process of the models. The table summarizes the average proof lengths across all models, revealing that the proof lengths range from 2.7 to 3.9. Notably, for both the Mathstral and Llama models, the enhanced versions consistently exhibit shorter proof lengths compared to their comb counterparts. This suggests that the models have learned to adopt more

Table 8: Performance per metric across iterations and tactic counts.

| Theorem Types | Tactics = 4 | | | Tactics = 8 | | | Tactics = 16 | | |
|---|---|---|---|---|---|---|---|---|---|
| | Round 1 | Round 2 | Total | Round 1 | Round 2 | Total | Round 1 | Round 2 | Total |
| # Candidate | 211,087 | 247,996 | 459,083 | 364,789 | 376,158 | 740,947 | 592,811 | 440,199 | 1,033,010 |
| # Deduplicated | 73,385 | 185,850 | 259,235 | 229,541 | 225,797 | 455,338 | 392,818 | 238,425 | 631,243 |
| # Correct | 16,296 | 69,765 | 86,061 | 52,220 | 132,791 | 185,011 | 68,771 | 135,796 | 204,567 |
| # Corrected | 6,395 | 13,912 | 20,307 | 21,916 | 24,147 | 46,063 | 31,306 | 24,593 | 55,899 |
| **Subtotal** | **22,691** | **83,677** | **106,368** | **74,136** | **156,938** | **231,074** | **100,077** | **160,389** | **260,466** |

efficient and simpler tactics for generating proofs through extensive training on large datasets. For instance, as shown in the example for test_087, Mathstral finetuned with LEANCOMB requires 11 steps to complete the proof, whereas Llama3 enhanced accomplishes the proof in just four steps.

```
theorem test_087_Mathstral_comb {n : ℕ} (hn : 0 < n):
  (ascPochhammer ℝ n).eval x = (x - 1 + n) * (ascPochhammer ℝ
    (n-1)).eval x := by
  unfold ascPochhammer
  cases n
  all_goals simp only [Polynomial.eval_one, CharP.cast_eq_zero,
    add_zero, mul_one]
  cases hn
  rename_i n
  cases n
  all_goals simp
  rename_i n
  rw [ascPochhammer_succ_right]
  simp only [Polynomial.eval_mul, Polynomial.eval_add,
    Polynomial.eval_X, Polynomial.eval_natCast]
  ring

theorem test_087_llama3_enhanced {n : ℕ} (hn : 0 < n):
  (ascPochhammer ℝ n).eval x = (x - 1 + n) * (ascPochhammer ℝ
    (n-1)).eval x := by
  have h1 : n = n - 1 + 1 := by
    rw [tsub_add_cancel_of_le]
    exact hn
  rw [h1, ascPochhammer_succ_right]
  simp only [Polynomial.eval_mul, Polynomial.eval_add,
    Polynomial.eval_X, Polynomial.eval_natCast,
    cast_add, cast_one, sub_add_add_cancel, add_tsub_cancel_right]
  rw [← mul_comm]
```

Beyond length reduction, enhanced models also show stronger proficiency in applying known algebraic identities and leveraging library theorems to simplify otherwise tedious transformations. For example, in test_048_1_dsv2_alp2, the model efficiently invokes symmetry properties of binomial coefficients and uses arithmetic reasoning (via omega) to resolve index equalities:

```
theorem test_048_1_dsv2_alp2 (n m k : ℕ)(hmk : m ≤ k) (hkn : k ≤
    n) :
    Nat.choose n k * Nat.choose k m = Nat.choose n (k - m) *
    Nat.choose (n - k + m) m := by
    rw [← choose_symm hmk]
    rw [choose_mul hkn (tsub_le_self)]
    rw [show k - (k - m) = m by omega]
    congr 2
    omega
```

Similarly, in test_074_1_dsv2_alp2, the model correctly applies the binomial expansion of $(x - 1)^n$ using the identity add_pow, and then constructs the appropriate sum transformation by reasoning over the sign alternation and index structure:

```
theorem test_074_1_dsv2_alp2 (x : ℝ)(n : ℕ) : (x - 1)^n=Σ k in
    Finset.range (n+1),Nat.choose n k * x ^ k * (-1 : ℝ ) ^ (n -
    k) := by
    rw [sub_eq_add_neg, add_pow]
    refine' sum_congr rfl fun k hk => _
    ring
```

Furthermore, we observed that after training on the enhanced dataset, models tend to rely heavily on the theorems from the LEANCOMB training set when generating proofs. However, despite this increased dependence on known theorems, the models occasionally introduce invalid or redundant steps in the proof process. For instance, in the following example, the step "lemma this" is an unnecessary operation; the proof remains valid even if this step is removed.

```
theorem test_031 (n : ℕ) :
  Σ k in Finset.range (n / 2 + 1), (-1 : ℝ) ^ k * Nat.choose (n -
    k ) k =(2 / Real.sqrt 3) *  Real.sin ((n + 1) * Real.pi / 3) :=
    by
  obtain h1 := Idt_32 n (-1 : ℝ)
  have h2 :
    Σ k in range (n / 2 + 1), ((n - k).choose k * (-1 : ℝ) ^ k) =
    Σ k in range (n / 2 + 1), ((-1) : ℝ) ^ k * choose (n - k) k :=
    by
      refine' sum_congr rfl fun k _ => _
      rw [mul_comm]
  rw [h2] at h1
  rw [h1]
  have : 1 + 4 * (-1 : ℝ) = -3 := by norm_num
  rw [this]
  exact_mod_cast complex_sqrt_neg n
```

## F.3 ANALYSIS OF THEOREM PROVING RESULTS ON PUTNAMBENCH

In this section, we examine the automated proof behavior of our models on the PUTNAMBENCH benchmark. The results reveal that the models are capable of constructing structurally sound and semantically accurate proofs, often emulating human-level strategies for algebraic manipulation and symbolic reasoning.

For example, in the case of putnam_1962_a5, the model successfully performs a sequence of algebraic rewrites and summation transformations to establish the equality between a closed-form expression and a finite sum involving binomial coefficients and powers:

```
theorem putnam_1962_a5
: ∀ n ≥ 2, (fun n : ℕ => (n * (n + 1) * 2^(n - 2) : ℕ → ℕ)) n = Σ
    k in Finset.Icc 1 n, Nat.choose n k * k^2 := by
    intro n hn
    rw [← Nat.Ico_succ_right (1 : ℕ) n]
    rw [← Nat.add_one]
    have h1 :  Σ k ∈ Finset.Ico 1 (n + 1), n.choose k * k ^ 2 = Σ k
      ∈ Finset.Ico 1 (n + 1), k ^ 2 * n.choose k := by
      refine' Finset.sum_congr rfl fun k hk => _
      rw [mul_comm]
    rw [h1]
    obtain h2 := idt_71' hn
    rw [h2]
    simp
```

Notably, the proof includes a nontrivial index transformation and applies a known identity (idt_71')
to simplify the summation, demonstrating the model's understanding of combinatorial symmetry.

Similarly, in the proof of putnam_1986_a1, the model effectively uses logical reasoning and inequality
manipulation to prove that a function defined on a constrained domain achieves its maximum value at
a specific point. The proof correctly applies numerical reasoning (norm_num), functional rewriting,
and nonlinear arithmetic tactics:

```
theorem putnam_1986_a1
    (S : Set ℝ) (f : ℝ → ℝ)
    (hS : S = {x : ℝ | x ^ 4 + 36 ≤ 13 * x ^ 2})
    (hf : f = fun x ↦ x ^ 3 - 3 * x) :
    IsGreatest
    {f x | x ∈ S}
    ((18) : ℝ )  := by
    simp only [hS, hf, Set.mem_setOf_eq, Set.mem_setOf_eq]
    refine ⟨?_, fun x hx ↦ ?_⟩
    refine ⟨3, by norm_num, by ring⟩
    obtain ⟨y, hy₁, rfl⟩ := hx
    nlinarith [sq_nonneg (y ^ 2 - 9), (by nlinarith : (0 : ℝ) ≤
    9)]
```

These examples highlight the model's ability to synthesize relevant lemmas, manipulate algebraic
structures, and apply inequalities effectively. However, occasional superfluous constructs—such as
unused hypotheses or redundant rewrites—can still appear, indicating room for refinement in proof
planning and step minimization.

## F.4 Analysis of Theorem Proving Results on CombiBench

We now turn our attention to the performance of the models on CombiBench, a suite focused on
combinatorial identities and discrete function transformations. These tasks often require models to
manipulate indexed sums, alternating signs, and recursive operators such as finite differences.

A representative example is the proof of a well-known identity involving the $k$-th forward difference
operator applied to a function $h : \mathbb{N} \to \mathbb{Z}$. The goal is to show that this $k$-fold operator evaluates to a
specific alternating sum involving binomial coefficients. Two variants of the proof below illustrate
different strategies employed by the models:

```
theorem brualdi_ch8_9 (h : ℕ → ℤ) (k n : ℕ): (fwdDiff 1)^[k] h n
    = Σ j ∈ Finset.range (k + 1),
    (-1 : ℤ) ^ (k - j) * Nat.choose k j * h (n + j) := by
    induction' k with k hk
    simp
    rw [fwdDiff_iter_eq_sum_shift]
    simp
```

In the first version, the model initiates an induction on k, aligning with the standard approach to
proving identities involving recursive operators. It simplifies the base case and applies the identity
fwdDiff_iter_eq_sum_shift to complete the inductive step. This structured approach reflects a strong
grasp of both the recursive nature of forward differences and the associated summation identities.

In contrast, the second version omits the inductive argument entirely:

```
theorem brualdi_ch8_9 (h : ℕ → ℤ) (k n : ℕ): (fwdDiff 1)^[k] h n
    = Σ j ∈ Finset.range (k + 1),
    (-1 : ℤ) ^ (k - j) * Nat.choose k j * h (n + j) := by
    rw [fwdDiff_iter_eq_sum_shift]
    simp
```

This version relies directly on applying the known identity fwdDiff_iter_eq_sum_shift, followed by
simplification. While this proof is shorter and still valid, it assumes the identity has already been

established or imported from prior training, bypassing the deeper structure that an inductive argument would expose.

These examples highlight an important trade-off: the more concise proof indicates effective pattern recognition and memorization of known identities, while the inductive version demonstrates constructive reasoning and generalizability. Across the benchmark, both behaviors were observed, with models choosing between efficiency and explicit derivation depending on the problem context.

# G   EXPERIMENTS DETAILS

In our experiment, we selected multiple large language models as baselines, covering two categories: one is general reasoning models, including Mathstral-8B (Mistral AI, 2024), LLaMA3-8B (Grattafiori et al., 2024), and Mistral-7B (Jiang et al., 2023); the other is models optimized for theorem proving, such as InternLM2.5-StepProver (Wu et al., 2024b) and DeepSeek-Prover-v2 (Guo et al., 2025). For step-by-step models, the model generates 16 candidate tactics for each proof step, which are deduplicated and ranked by their log-likelihood values. In the whole proof method, we call the model 32 times per Pass1, and a proof is considered successful if it passes once. All proofs were verified using Lean 4 (v4.14.0) and the corresponding Mathlib4.

## G.1   DETAILS AND HYPERPARAMETERS

We employed Llama-3.1-8B-Instruct as the candidate tactic model within our generator ATG4CI. This model is a transformer-based, autoregressive language model optimized for high performance. To enhance its alignment with human preferences in terms of utility and safety, we fine-tuned the model using Supervised Fine-Tuning (SFT) (Wei et al., 2022) and Reinforcement Learning from Human Feedback (RLHF) (Ouyang et al., 2022). The reinforcement learning component includes a policy network and a critic network, both consisting of two linear layers with a hidden size of 16. The training spans 10 iterations. Each decision step performs 100 simulations, and each iteration includes 20 total episodes. Each node may generate up to 16 candidate theorems, with a search time limit of 600 seconds.

---

**Prompt Example:**

Complete the following Lean 4 code:
```lean4

[Statement]:
```

```
1  import Mathlib
2  import Aesop
3  import Theorem
4  set_option maxHeartbeats 0
5  open BigOperators Real Nat Topology Rat Finset
6
7  theorem test_074 (x : ℝ) (n : ℕ) :
8      (x - 1)^n = Σ k in Finset.range (n + 1), Nat.choose n k *
       x^k * (-1 : ℝ)^(n - k) := by
```

---

All training tasks were conducted on a cluster of six NVIDIA L40 GPUs (48GB each). For SFT, we employed Low-Rank Adaptation (LoRA) (Hu et al., 2021), leveraging bfloat16 mixed precision and DeepSpeed ZeRO Stage 0 (Jacobs et al., 2023) for memory optimization. LoRA was applied to all layers of the model. We used the AdamW optimizer (Dettmers et al., 2022) with a batch size of 4. The learning rate was linearly warmed up from 0 to $5.0 \times 10^{-5}$ over the initial $5\%$ of training steps, followed by a cosine decay schedule for subsequent updates.

We adopted a best-first search (BFS) (Pearl, 1984) tactic for evaluation. At each search step, the model generated 16 tactic candidates, deduplicated and ranked according to their log-likelihoods.

Each candidate was subsequently verified using Lean 4, determining whether it should have been discarded or expanded into new proof states.

The ranking of states is based on the cumulative log-likelihood of the tactics leading to the current state. The model is prompted using a structured (state, tactic) format, enabling the generation of tactics conditioned on the current proof state. The prompt examples are presented as follows:

[Output]:

```
theorem test_074 (x : ℝ) (n : ℕ) :
    (x - 1)^n = Σ k in Finset.range (n + 1), Nat.choose n k * x^k
    * (-1 : ℝ)^(n - k) := by
  rw [sub_eq_add_neg]
  rw [add_comm]
  rw [add_comm]
  -- Apply the binomial theorem to (x + (-1))^n
  rw [add_pow]
  -- Apply sum_congr to match the form of the binomial expansion
  apply sum_congr rfl
  intro k hk
  -- implify the terms using const_mul
  simp at hk
  have h : k ≤ n := by linarith
  simp [mul_assoc, mul_comm, mul_left_comm, h]
```

**Prompt Example:**

You are using Lean 4 for theorem proving. You are proving a theorem in Lean 4. Based on the current state of the theorem, provide the most reasonable proof tactic. Ensure your tactic is syntactically correct according to Lean 4's tactic syntax and effectively progresses the proof.

[Current State]:
$n : \mathbb{N}$
$\vdash Nat.choose\,(2*n)\,n \leq \sum x \in range\,n,\ Nat.choose\,(2*n)\,x + Nat.choose\,(2*n)\,n$

[Output Tactic]:

       rw [sum_range_add], $-0.30987493962877327$

       rw [sum_range_add_choose] , $-0.3457271686162461$

       rw [sum_range_add_choose_add] , $-0.3690287694334984$

       rw [sum_range_add_choose_eq] , $-0.44889247231185436$

       ......

