# OpenReview forum: "LeanComb: A Combinatorial Identities Benchmark for Automated Theorem Proving"
_ICLR.cc/2026/Conference — ICLR 2026 Conference Withdrawn Submission_

### Official Review · Reviewer_gQus · 2025-10-31

**Soundness:** 3
**Presentation:** 1
**Contribution:** 3
**Rating:** 6
**Confidence:** 3

**Summary:**

This paper introduces the following contributions:
1. LeanComb, a formalized combinatorial identities in Lean 4 annoted by human, covering 8 core areas
2. ATG4CI, a framework to generate combinatorial identities automatically via the following pipeline (1) Extract the partial proof paths (P3s) from each theorem in the LeanComb training set (2) For each partial proof path, use a tactic generation LLM to generate n candidate tactics for each step to perform a PUCT proof tree search to find some candidate theorems. (3) Deduplicate and correct the candidate theorems.
3. LeanComb++, a dataset generated by ATG4CI that includes 260K+ verified formal combinatorial theorems. The authors show that models fine-tuned on LeanComb++ can obtain significant performance gain on related benchamrks.
4. Lean4kit, a toolkit that can extract all state-tactic pairs from any repository in Lean4, for assisting LLM theorem provers to interact with Lean.

**Strengths:**

- This work fills in the blank that there is no enough combinatorics formal data in public.
- The performance gain through fine-tuning the LLMs on LeanComb++ is impressive on tree-search methods (~12% to ~24%).
- The efforts on recovering a correct proof from a failed candidate theorem derived from a partial proof path are very comprehensive.

**Weaknesses:**

1. Lack of enough background introduction for readers who are unfamiliar with combinatorics or Lean. Since the contents of the paper are substantial, it becomes difficult for the authors to make detailed introduction on combinatorics and Lean basics in the main body.

2. The description of ATG4CI is too coarse to let readers precisely understand. (1) During describing the candidate theorem generation process, the authors say "Q(s, t) is the estimated value of the state-tactic pair, obtained from the value network or learned from past simulations". However, throughout this paper, it is not mentioned which approach they take in practice. Besides, the authors claim that they use a critic model $C_\theta$, which is no longer mentioned anywhere. (2) For the theorem correction process, the authors say "Those that fail verification are categorized by error type and corrected using the corresponding correction methods", but the readers are not referred to appendix C.3 where the correction methods are introduced, which has confused me for a long while.

3. Training on LeanComb++ seems to degrade the performance of DeepSeek-Prover-V2-7B when using the whole-proof generation method, which is the mainstream method for SOTA theorem provers.

**Questions:**

1. The CT generation process only includes the leaf nodes. Why? Won't this make the average difficulty of the LeanComb++ problems easier than expected (since these problems are expected to be solved by a single tactic or one-step simple Incomplete/Type/Logical correction)?

2. How do you think about the overall quality of the LeanComb++ problems? Since they are all transformed from the root theorems, I expect that they are quite similar to the corresponding root theorems and require similar proof techniques. What if we want to acquire training data beyond the root theorems and the tactics known by the LLM?

3. The tactic generation model is rewarded to complete the proof, while ATG requires more diverse candidate theorems. Can you come up with some techniques to solve this issue naturally?

---

### Official Review · Reviewer_sPFh · 2025-11-01

**Soundness:** 2
**Presentation:** 3
**Contribution:** 2
**Rating:** 4
**Confidence:** 4

**Summary:**

This paper proposes a new dataset for formal theorem proving that focuses on combinatorial identities. The dataset includes 727 theorems in total and is further augmented by extracting subgoals from the seed theorems. The paper designs a reinforcement learning pipeline to iteratively train LLaMA3.1-8B to produce partial proofs and transforming the remaining subgoals into new theorems. A family of LLM provers at the scale of ~8B solve several more problems in CombiBench and PutnamBench after finetuning on the designed dataset.

**Strengths:**

The paper has contributed to alleviating the scarcity of formalized math statements by expert-annotated statements and proofs, which costs 1800 hours of work. The statements are verified by provided full proofs and are deduplicated, which offers a much more cleaner training dataset for neural provers compared with larger-scale datasets such as Lean Workbook. The validity of the training dataset is demonstrated by increased performance of small-scale LLMs on standard benchmarks.

**Weaknesses:**

1. The paper is motivated from the scarcity of combinatorics theorems in existing datasets. However, the proportion of combinatorics statements (93/627) is still relatively small in the proposed dataset, according to Figure 1. In contrast, Mathlib4 contains 2K combinatorics related statements.

2. It's unclear how the dataset supplements CombiBench for combinatorics. It will be more convincing to show the different ranks among SOTA provers between the proposed benchmark and standard benchmarks.

3. The subgoal extraction procedure shares a similar idea from Goedel-prover-v2. However, subgoal extraction only provides easier problems than the seed theorem. In contrast, Goedel-prover also employs LLMs to generate harder variants.

4. The evaluation is conduced on 8B scale models, which shows several more solved problems on PutnamBench. The maximal solved number is 10. In fact, the SOTA neural provers (seed prover) solve over 300 problems on PutnamBench. MiniF2F is a better benchmark for small-scale provers.

**Questions:**

Please refer to the weaknesses.

---

### Official Review · Reviewer_GmS3 · 2025-11-07

**Soundness:** 2
**Presentation:** 2
**Contribution:** 2
**Rating:** 2
**Confidence:** 4

**Summary:**

This paper presents a human-curated Lean 4 dataset of combinatorial identities across multiple subareas, and an automated theorem-generation pipeline that mines theorems from partial proof paths using LLM-proposed tactics guided by PUCT search.

**Strengths:**

This Lean4 combinatorics dataset fills a gap in the Lean ecosystem by delivering a domain-specific resource for an underrepresented area.

**Weaknesses:**

The augmentation pipeline bootstraps from a small, human-curated set of seed theorems and mines partial proof states along those seeds’ proof trees. Because failed branches are routinely turned into “new” theorems by reusing (or conditioning on) the seed statement, many outputs are very likely to be close restatements or conditional variants of the seeds rather than independent and meaningful identities. As a result, the large scale augmented dataset is very likely to lack semantic diversity, and the augmentation process can degenerate into producing near-tautological statements that offer little new mathematical content.  Also, the pipeline lacks novelty since it mainly recombines well-established ingredients like state–tactic mining, PUCT-style search, and promoting intermediate/failed goals to conjectures.

**Questions:**

How do you detect semantic duplication from your data with Mathlib, CombiBench, etc.?

---

### Official Review · Reviewer_SdhT · 2025-11-10

**Soundness:** 2
**Presentation:** 3
**Contribution:** 2
**Rating:** 4
**Confidence:** 3

**Summary:**

This paper introduces LEANCOMB, a manually curated Lean 4 dataset of 418 combinatorial identities and 209 supporting lemmas, along with ATG4CI, a “learn-from-failure” automated theorem generation pipeline that converts partial failed proof paths into valid new theorems. Using ATG4CI, the authors construct LEANCOMB++, a large synthetic dataset of 260k automatically generated combinatorics theorems with Lean-verified proofs. Experiments show that fine-tuning several LLM-based theorem provers on LEANCOMB++ improves pass@k performance on the LEANCOMB test set and provides modest gains on CombiBench and the combinatorics subset of PutnamBench. Ablation studies compare the LLM+PUCT generator to simpler MCTS/BFS baselines, demonstrating that the proposed pipeline produces more useful and diverse theorems.

**Strengths:**

- LEANCOMB is carefully curated with substantial human effort (1,800+ hours) and clear annotation protocols. The dataset fills a gap in combinatorics identities.
- The ATG4CI pipeline is conceptually interesting: transforming failed partial proofs into nearby true theorems is elegant and appears effective in this algebraically structured domain.
- LEANCOMB++ (~260k theorems) is impressive, and the authors provide thorough analyses of theorem diversity, proof lengths, and tactic distributions.
- Models fine-tuned on LEANCOMB++ consistently improve pass@k on the LEANCOMB test set (+7.2% on average), and some improvements transfer to CombiBench and PutnamBench (combinatorics subset).
- The comparisons of ATG4CI vs naive MCTS vs BFS demonstrate that LLM+PUCT generates fewer but more useful theorems—an important finding for ATG research.

**Weaknesses:**

- The paper uses the term “ATP” in a non-standard way. In classical automated reasoning, ATP refers to symbolic automation systems—such as Sledgehammer in Isabelle or Aesop in Lean 4. The models in this paper are Neural Theorem Provers (NTP), i.e., LLM-based proof generators whose outputs are only validated by Lean, so calling them ATP may mislead readers about the nature and scope of the contribution.
- The paper does not evaluate on MiniF2F, the standard cross-domain benchmark in formal mathematics. MiniF2F also provides fine-grained categories, which would help identify whether the synthetic data improves algebraic manipulation only or transfers to broader reasoning abilities.
- The value of the large synthetic dataset LEANCOMB++ is not fully quantified. There are no dataset-size ablations, so it is unclear whether the full 260k theorems are necessary or whether a much smaller subset would achieve similar improvements.
- Some citations and terminology choices are incorrect or imprecise, such as citing LeanWorkbook for “Lean 4” and using ATP terminology inconsistently. These issues affect clarity and make the relationship to prior work less precise than it could be.

**Questions:**

- MiniF2F evaluation:
Can the authors provide results on MiniF2F, ideally with category-level breakdowns? This would clarify whether the proposed data augmentation improves general mathematical reasoning or mainly benefits combinatorics-related algebraic identities.
- Dataset-size ablations:
How sensitive are the results to the scale of LEANCOMB++? It would be helpful to see performance when training on smaller subsets (e.g., 25%, 50%) to understand whether the full 260k generated theorems are necessary.

---

### Note · Authors · 2025-11-12

I have read and agree with the venue's withdrawal policy on behalf of myself and my co-authors.